# Evaluation of Methanotroph (*Methylococcus capsulatus*, Bath) Bacteria Protein as an Alternative to Fish Meal in the Diet of Juvenile American Eel (*Anguilla rostrata*)

**DOI:** 10.3390/ani13040681

**Published:** 2023-02-15

**Authors:** Wenqi Lu, Haixia Yu, Ying Liang, Shaowei Zhai

**Affiliations:** 1Engineering Research Center of the Modern Industry Technology for Eel, Ministry of Education, Fisheries College of Jimei University, Xiamen 361021, China; 2Key Laboratory of Healthy Mariculture for the East China Sea, Ministry of Agriculture and Rural Affairs, Jimei University, Xiamen 361021, China

**Keywords:** methanotroph bacteria protein, growth, serum immunity, intestinal health, intestinal microbiota, *Anguilla rostrata*

## Abstract

**Simple Summary:**

It is an urgent issue to reduce the consumption of high-quality fish meal for the sustainable development of eel aquaculture. Methanotroph (*Methylococcus capsulatus*, Bath) bacteria protein is shown to be a promising alternative to fish meal due to its high nutritional values and lower production costs. In the present study, methanotroph bacteria protein was used to substitute the fish meal in the diet of the American eel (*Anguilla rostrata*); it could replace 6% of fish meal without negatively affecting on growth and health status of American eels. This study provides a first perspective on the feasibility of replacing fish meal with bacteria protein in the diet of the American eel.

**Abstract:**

This study was conducted to evaluate the effects of replacing fish meal (FM) with methanotroph (*Methylococcus capsulatus*, Bath) bacteria protein (MBP) in the diets of the juvenile American eel (*Anguilla rostrata*). Trial fish were randomly divided into the MBP0 group, MBP6 group, MBP12 group, and MBP18 group fed the diets with MBP replacing FM at levels of 0, 6%, 12%, and 18%, respectively. The trial lasted for ten weeks. There were no significant differences in weight gain or feed utilization among the MBP0, MBP6, and MBP12 groups (except for the feeding rate in the MBP12 group). Compared with the MBP0 group, the D-lactate level and diamine oxidase activity in the serum were significantly elevated in the MBP12 and MBP18 groups. In terms of non-specific immunity parameters in serum, the alkaline phosphatase activity was significantly decreased in the MBP18 group, and the complement 3 level was significantly elevated in the MBP12 and MBP18 groups. The activities of lipase and protease in the intestine were significantly decreased in the MBP12 and MBP18 groups. Compared with the MBP0 group, the total antioxidant capacity and activities of superoxide dismutase, catalase, and glutathione peroxidase in the intestine were significantly decreased in the MBP18 group, while the malondialdehyde level was significantly increased. The villus height, muscular thickness, and microvillus density were significantly decreased in the MBP12 and MBP18 groups. There were no significant differences in the foresaid parameters between the MBP0 group and the MBP6 group. The intestinal microbiota of the MBP6 group was beneficially regulated to maintain similar growth and health status with the MBP0 group. The adverse effects on the intestinal microbiota were reflected in the MBP18 group. In conclusion, MBP could successfully replace 6% of FM in the diet without adversely affecting the growth performance, serum biochemical parameters, and intestinal health of juvenile American eels.

## 1. Introduction

Fish meal (FM) is recognized as an excellent protein source in aquatic feeds due to its ideal amino acid ratio, balanced nutrition, high digestibility, and good palatability [1,2]. In recent years, with the rapid expansion of the aquaculture industry, there has been an increasing demand for FM coupled with a significant shortage in global FM production, resulting in intense competition for its use by the aquatic animal feed industry [3]. Therefore, the exploring of alternative proteins as partial or total replacements for FM has become a strong focus for the aquaculture industry. One particular group of alternative proteins are single cell proteins (SCPs), which are protein concentrates produced by non-pathogenic microorganisms, such as bacteria, fungi, microalgae, and yeast using waste gas, waste water, or waste residue for large-scale culture [4,5]. SCPs not only have high nutritional value but also have the advantages of high production efficiency, lower price, abundant sources of raw materials, and being environmentally friendly [6].

As one of the bacterial proteins, methanotroph (*Methylococcus capsulatus*, Bath) bacteria protein (MBP) is mainly developed from methanotrophs, which are gram-negative bacteria that use methane as their sole source of carbon and energy. The abundant supply, cheap transportation, and reasonable cost of methane from natural gas indicate that MBP production from methane fermentation is economical at a large scale, and MBP might be the most promising bacterial protein [7,8]. MBP is high in protein content, similar in amino acid composition to FM, and rich in vitamins and minerals [9,10]. Given its nutritional properties, several studies have reported that MBP could partially replace FM without inhibiting the growth performance of Atlantic halibut (*Hippoglossus hippoglossus*) [11], Atlantic salmon (*Salmo salar*) [10,12], Japanese yellowtail (*Seriola quinqueradiata*) [1], largemouth bass (*Micropterus salmoides*) [13], rainbow trout (*Oncorhynchus mykiss*) [14,15,16], and turbot (*Scophthalmus maximus* L.) [17]. Furthermore, the emerging evidence suggested that MBP might have beneficial effects on the antioxidant capacity, immunity, and intestinal health of some fish species [8,18,19,20,21]. The effectiveness, impact, and appropriate inclusion levels of MBP in many other fish species globally would need to be confirmed.

Known as “ginseng in water”, eel (*Anguilla* spp.) is one of the most common fish species cultured around the world, and it plays an important role in international trade. The American eel (*Anguilla rostrata*) is one of the main cultured eel species in China since the European eel (*Anguilla anguilla*) has been listed as an extremely endangered species [22]. As a carnivorous fish, it has a high requirement for dietary protein, and the highest quality FM is often used as the main protein source in the feed [23]. At present, no study has been conducted to use MPB to replace FM in eel diets. Therefore, the present study aimed to evaluate the potential effects of MBP as a substitute for the FM on growth performance, serum biochemical parameters, and intestinal health of juvenile American eels.

## 2. Materials and Methods

### 2.1. Trial Design and Diets

A total of 480 American eels (26.99 ± 0.03 g/fish) were randomly divided into four treatment groups, which were the MBP0 group, MBP6 group, MBP12 group, and MBP18 group, respectively. There were four replicates in each treatment group with 30 eels per replicate. The four groups were fed the four isonitrogenous (about 49% of crude protein) and isolipidic (about 6% of crude lipid) diets with MBP replacing FM at levels of 0, 6%, 12%, and 18%, respectively. The levels of FM replaced with MBP in the present trial were based on the optimal MBP levels in the diets of other carnivorous fish species in previous reports [11,13,16,21]. The composition and nutrient levels of the four diets are shown in Table 1. The amino acid levels of the trial diets are presented in Table 2. For diet preparation, each ingredient was crushed using a ZFJ-300 feed grinder (Jiangyin Ruizong Machinery Manufacturing Co., Ltd., Wuxi, China) and then passed through an 80 µm mesh sieve before being mixed well and stored at −20 °C until required.

### 2.2. Feeding Trial and Management

Six hundred juvenile American eels were provided by a fishery company (Fujian Jinjiangzhiman Aquatic Technology Co., Ltd., Zhangzhou, China). To acclimate the trial conditions, eels were kept in a circular tank (800 L water, diameter × height: 160 cm × 110 cm) with a water recirculation system (water change rate: 5 L/min) for four weeks. All eels were fed on a commercial diet manufactured by Fuzhou Xinruiyi Industrial Co., Ltd., Fuzhou, China. The proximate composition of the commercial diet was 46.28% crude protein, 7.35% crude lipid, 5.14% moisture, and 14.66% ash. Eels were fed to apparent satiation twice daily (6:00 a.m. and 6:00 p.m.). During each feeding, the powder diet was formed into a dough shape (powder diet mixed with 1:1.1 volume water) and placed on a feeding table for eels. After 30 min, the uneaten feed was carefully collected with a net, dried, and weighed to calculate the feed consumption. The number and weight of dead eels were recorded daily. Before the formal trial, all the fish were starved for 24 h. After four weeks of acclimation, 480 healthy American eels of similar size were selected and randomly distributed into sixteen circular PVC tanks (approximately 300 L of water) with a water recirculation system. Three days were taken to convert the commercial diet to the trial diet gradually. During the formal trial period, fish management was the same as that during the adaption period. The water quality was maintained as follows: water temperature 25 ± 1 °C, pH 7.25 ± 0.25, nitrite 0.045 ± 0.015 mg/L, total ammonia nitrogen 0.4 ± 0.2 mg/L, and dissolved oxygen 7.4 ± 0.6 mg/L. The formal trial lasted for ten weeks.

### 2.3. Sample Collection

At the termination of the feeding trial, all eels were starved for 24 h and then were anesthetized with eugenol (100 mg/L) [24]. The number and weight of eels per tank were measured for calculating the parameters of growth performance. The blood of twelve eels from each tank was collected, centrifuged (10,000 r/min, 10 min) at 4 °C, and pooled for determining serum biochemical indexes according to the procedures described by Liu et al. [25]. After that, intestine samples of six bled eels from each tank were sampled, homogenized (with 0.86% normal saline), centrifuged (3000 r/min, 10 min) at 4 °C, and stored (−80 °C) for measuring activities of intestinal digestive and antioxidant enzymes. Midguts were sampled from three bled eels in each tank and immersed in Bouin’s solution for morphological observation. The aforementioned three midguts from each tank were taken, fixed, and stored at 4 °C for scanning electron microscopy examination. Additionally, six intact intestine tissue samples per tank were collected and frozen at −80 °C for analyzing intestinal microbiota.

### 2.4. Analysis Methods

#### 2.4.1. Calculations of Growth Performance Parameters

Growth performance parameters of survival rate (SR), weight gain rate (WGR), specific growth rate (SGR), feeding rate (FR), feed efficiency (FE), and protein efficiency ratio (PER) were calculated as follows:SR (%) = (final fish numbers per tank/initial fish numbers per tank) × 100; 
WGR (%) = [final fish weight per tank (g) − initial fish weight per tank (g)]/initial fish weight per tank (g) × 100;
SGR (%/d) = [(Ln final fish weight per tank (g) − Ln initial fish weight per tank (g)) × 100]/days;
FR (%) = feed consumption per tank (g)/[(initial fish weight per tank (g) + final fish weight per tank (g))/2]/days;
FE (%) = [final fish weight per tank (g) − initial fish weight per tank (g)]/feed intake per tank (g) × 100;
PER (%) = [final fish weight per tank (g) − initial fish weight per tank (g)]/[feed consumption per tank (g) × protein content of the diet] × 100.

#### 2.4.2. Proximate Analysis and Amino Acid Levels of Trial Diets

The proximate composition of trial diets was determined according to standard methods of AOAC [26]. Briefly, the moisture content was determined by drying the samples in an oven to constant weight. The crude protein content was determined using the Kjeldahl method (N × 6.25, 2300-Kjeldahl apparatus, FOSS, Denmark) after acid digestion. The crude lipid content was determined using the Soxhlet method (36680-analyzer, BUCHI, Flawil, Switzerland) after ether extraction. The ash content was determined by burning samples in a muffle furnace at 550 °C for 5 h. The gross energy was determined using an adiabatic bomb calorimeter (Model 1281, PARR, Moline, IL, USA).

The amino acid levels of trial diets (Table 2) were measured by the Feed Industry Center of China Agricultural University (Beijing, China). Briefly, the samples were lyophilized using a freeze dryer first, and then 17 amino acids (except for tryptophan) were analyzed by an automatic amino acid analyzer (L-8900, Hitachi, Tokyo, Japan) following hydrolysis in 6 mol/L HCl at 110 °C for 24 h. Methionine and cysteine were determined using formic acid (88% formic acid: 30% hydrogen peroxide = 9:1) to protect before acid hydrolysis (6 mol/L HCl). Tryptophan was analyzed using the alkaline hydrolysis method by a reverse-phase HPLC (LC10, Shimadzu, Kyoto, Japan).

#### 2.4.3. Serum Biochemical Analysis

Assay kits (Nanjing Jiancheng Bioengineering Institute, Nanjing, China) were selected to determine serum biochemical parameters, including glutamic-oxalacetic transaminase (GOT, C010-2-1), glutamic-pyruvic transaminase (GPT, C009-2-1), D-lactate (D-lac, H263-1-2), lysozyme (LZM, A050-1-1), acid phosphatase (ACP, A060-2), alkaline phosphatase (AKP, A059-2), and complement 3 (C3, H186-1-2). The diamine oxidase (DAO, A088-1) activity was measured using an ELISA kit (Shanghai Jianglai Biotechnology Co., Ltd., Shanghai, China).

#### 2.4.4. Activities of Intestinal Digestive Enzymes and Antioxidant Parameters

Activities of intestinal digestive enzymes, including amylase (C016-1-1), lipase (A054-2-1), and protease (A080-3-1) were detected by commercial kits (Nanjing Jiancheng Bioengineering Institute, Nanjing, China) according to the instructions.

The intestinal antioxidant parameters, including total antioxidant capacity (T-AOC, A015-2-1), superoxide dismutase (SOD, A001-1), catalase (CAT, A007-1-1), glutathione peroxidase (GSH-Px, A005-1), and malondialdehyde (MDA, A003-1) were tested according to the assay kit instructions (Nanjing Jiancheng Bioengineering Institute, Nanjing, China).

#### 2.4.5. Intestinal Morphology

Intestinal tissue sections were prepared following the method described by Chen et al. [24]. The sections were stained with hematoxylin and eosin (H&E) and examined with a positive fluorescence microscope (BX80-JPA, Olympus, Tokyo, Japan) to take representative photomicrographs. Image-Pro Plus 6.0 software (Media Cybernetics, Silver Spring, MD, USA) was used for morphometric analysis. The methods to measure the intestinal villus height and muscular thickness were also the same as the study of Chen et al. [24].

Samples of intestines for scanning electron microscopy were treated as follows: midgut samples were fixed in glutaraldehyde for a minimum of 24 h before being cleaned with phosphoric acid buffer (0.1 M) three times, and then dehydrated with a graded series of ethanol. The treated samples were dried in a critical point dryer (K850, Quorum, Sussex, UK), and then gold coated in an ion sputtering instrument (MC1000, Hitachi, Tokyo, Japan), before being observed and photographed on a scanning electron microscope (SU8100, Hitachi, Tokyo, Japan).

#### 2.4.6. Intestinal Microbiota Analysis

Considering no consistent changes in growth performance, serum biochemical parameters, and some intestinal health indexes in the MBP12 group, the intestine samples from the MBP6 and MBP18 groups were chosen for intestinal microbiota analysis in comparison with the MBP0 group. The high-throughput sequencing of intestinal microbiota was performed by the Illumina Miseq PE300 platform (Beijing Allwegene Tech. Co., Ltd., Beijing, China). The procedure of intestinal microbiota profiling followed the specific methods described by Shi et al. [27].

### 2.5. Statistical Analysis

All trial data were expressed as mean ± SD (standard deviation, *n* = 4) and analyzed by one-way ANOVA model using SPSS 20.0 analysis software (IBM, Armonk, NY, USA). Duncan’s test was used for multiple comparisons if significant differences existed. *p* < 0.05 indicated a statistically significant difference. When data were expressed as percentages, squared arcsine transformation was performed before statistical analysis. For intestinal microbiota analysis, the alpha diversity index was analyzed using QIIME (v1.8.0, http://bio.cug.edu.cn/qiime/ accessed on 24 October 2021) software and Lefse analysis was performed using R-Statistical software v3.6.0 (R Statistical Corp., Vienna, Austria). The Linear Discriminant Analysis (LDA) score screening value was set to 3, *p* < 0.05.

## 3. Results

### 3.1. Growth Performance

The growth performance parameters of juvenile American eels are shown in Table 3. The SR of juvenile American eels was not significantly influenced by dietary MBP levels (*p* > 0.05). Although all post-treatment growth performance parameters were significantly decreased in the MBP18 group (*p* < 0.05), no significant differences were found between the growth parameters for the MBP0, MBP6, or MBP12 groups (*p* > 0.05), except for the FR of the MBP12 group, which was significantly reduced compared with the MBP0 group (*p* < 0.05). However, there was a decreasing trend of WGR in the MBP12 group in comparison with that of the MBP0 group (*p* = 0.14).

### 3.2. Serum Biochemical Parameters

As shown in Table 4, activities of GOT and GPT were significantly increased by dietary MBP levels (*p* < 0.05) compared with the MBP0 group, and the highest value for GPT was observed in the MBP18 group. Compared with the MBP0 group, the D-lac level and DAO activity were significantly increased in MBP12 and MBP18 groups (*p* < 0.05), and the highest values of those two parameters were found in the MBP18 group.

In terms of non-specific immunity, there was no significant difference in activities of LZM or ACP among all the groups (*p* > 0.05), although there was a decreasing trend in LZM with increasing MBP inclusion. The AKP activity was significantly decreased in the MBP18 group (*p* < 0.05), and no significant differences were found among the other three groups (*p* > 0.05). Compared with the MBP0 group, the C3 level was significantly decreased in MBP12 and MBP18 groups (*p* < 0.05), with the lowest value being noted in the MBP18 group.

### 3.3. Intestinal Digestive Enzymes

Activities of intestinal digestive enzymes are presented in Table 5. No significant difference in amylase activity was found among all the groups (*p* > 0.05). Compared with the MBP0 group, the lipase and protease activity were significantly decreased in the MBP12 and MBP18 groups (*p* < 0.05), and the lowest values were observed in the MBP18 group.

### 3.4. Intestinal Antioxidant Parameters

As shown in Table 6, compared with the MBP0 group, the T-AOC level and activities of SOD, CAT, and GSH-Px were significantly decreased only in the MBP18 group (*p* < 0.05), and no significant differences were found among the other three groups (*p* > 0.05). The content of MDA was increased only in the MBP18 group (*p* < 0.05).

### 3.5. Intestinal Morphology

As shown in Figure 1 and Table 7, the villus height and muscular thickness of the intestine in the MBP12 and MBP18 groups were significantly lower than those in MBP0 and MBP6 groups (*p* < 0.05), and the lowest values were both observed in the MBP18 group. As evidenced in the SEM view, the intestinal microvillus density in MBP12 and MBP18 groups was sparser than those in MBP0 and MBP6 groups, and there was certain damage to the intestinal microvillus in the MBP18 group (Figure 2).

### 3.6. Intestinal Microbiota

As shown in Figure 3, although there were no significant differences in richness indexes (Chao 1 and Observed_species) or diversity (PD_whole_tree and Shannon) among all the groups (*p* > 0.05), there was an increasing trend in these measures of the MBP18 group compared with the MBP0 group. At the phylum level (Figure 4), Firmicutes, Proteobacteria, Actinobacteriota, and Bacteroidota were the four predominant bacteria in MBP0, MBP6, and MBP18 groups. Compared with the MBP0 group, the relative abundance of Firmicutes in the MBP6 group was increased, while the relative abundances of Proteobacteria, Actinobacteriota, and Bacteroidota in the MBP6 group were decreased. No obvious differences in the relative abundance of Firmicutes or Proteobacteria were found between the MBP0 and MBP18 groups. The relative abundance of Actinobacteriota in the MBP18 group was lower than that in the MBP0 group, while the relative abundance of Bacteroidota in the MBP18 group was higher than that in the MBP0 group.

The Lefse analysis in intestinal microflora of American eel based on genus level is shown in Figure 5. A total of 14 bacterial species were noted with significant differences among the MBP0, MBP6, and MBP18 groups (*p* < 0.05). Higher relative abundances of *Chryseobacterium*, *Leucobacter*, and *Microbacterium* were found in the MBP0 group, while higher relative abundances of *Enterococcus*, *Macrococcus*, *Staphylococcus*, *Clostridium-sensu-stricto*, *Aeromonas*, and *Schlegelella* were observed in the MBP6 group, and higher relative abundances of *Bosea*, *Rhodobacter*, *Prevotellaceae_NK3B31_group*, *Pedomicrobium*, and *Brevundimonas* were shown in the MBP18 group.

## 4. Discussion

### 4.1. Effects of FM being Replaced by MBP on Growth Performance of Juvenile American Eel

The present study showed that the juvenile American eels were readily able to tolerate 6% MBP in diets without negative effects on a range of growth measures. However, the WGR, SGR, FR, FE, and PER showed a decreasing trend when the replacement levels were up to 12% or 18%, although there were no statistical differences (*p* = 0.14) between the MBP0 and MBP12 groups. This suggested that MBP could successfully replace 6% of the FM in the diet of the juvenile American eel. Previous studies showed that when the inclusion levels of MBP were similar to the present study in diets of hybrid grouper (*Epinephelus fuscoguttatus* ♀ × *E. lanceolatus* ♂) at 8% MBP [4], in tilapia (*Oreochromis niloticus*) diets at 8.5% MBP [18], and in Atlantic halibut diets at 9% MBP [11], there was no negative impact on growth or feed utilization. In contrast to the results obtained in the present study for the MBP6 group, higher inclusion levels of MBP were reported in largemouth bass (*Micropterus salmoides*) diets at 12.9% MBP [13], in Japanese yellowtail at 17% MBP [1], and in Atlantic salmon at 20% MBP [12] with no obvious adverse effect on the growth of those fish species. Due to differences in quality and addition proportion of FM, diet formulation, and study design, it is not clear why different fish species have different upper limits for MBP inclusions in their diets. Typically, there is a concomitant decrease in feed intake when approaching or exceeding the upper inclusion limit which translates into a reduction in weight gain or growth. It is known that fish utilize a number of olfactory and gustatory cues that drive feed intake, partly driven by FM in the diet. It is possible that reductions in FM impact on feeding behaviors, particularly at higher MBP inclusion levels, although this will be species dependent. Understanding the interplay between FM and MBP levels in the diet and growth outcomes requires further study in the future.

In addition, it should be noted that the values of WGR and SGR of American eels in the present study were relatively lower than those of other fish species. This phenomenon might be related to the differences in body size of trial fish species. Eels with bigger body sizes usually presented a lower growth rate, which could be confirmed in previous studies [24,28,29].

### 4.2. Effects of FM Replaced by MBP on Serum Biochemical Parameters of Juvenile American Eel

Some important physiological functions and health status of fish are often reflected by serum biochemical parameters [30], including serum GOT and GPT, which can reflect the degree of liver cell damage [31,32]. In the present study, with the increase in FM replacement level by MBP, activities of GPT and GOT in the serum were significantly increased, which indicated that the higher inclusion level of MBP could damage the liver of American eels. Similar to the present result, the FM replacement with a high inclusion level of MBP increased the GPT activity in the serum of Japanese yellowtail [1]. However, Zhang et al. [13] reported that activities of GOT and GPT in the serum of largemouth bass were not affected by dietary MBP levels.

D-lactate and DAO in the serum are the markers of intestinal permeability. The increased levels of these parameters indicated that there might be some damage to the intestinal mucosal barrier [33]. Elevated levels of D-lactate in rainbow trout and DAO in Japanese seabass (*Lateolabrax japonicus*) with increased levels of dietary soybean suggested that there was an increase in membrane permeability and damage to intestinal mucosal cells [34,35]. In the present study, there was a slight reduction in D-lactate level and DAO activity in the serum in the MBP6 group compared with the MBP0 group, which suggested that 6% MBP inclusion in the diet could not impair the barrier function. The significant increase in those parameters in the 12% and 18% MBP groups indicated that there might be some impairment of the intestinal mucosal barrier function of juvenile American eels, which might be supported by the reductions in villus height and muscular thickness, as shown in Figure 1 and Table 7.

ACP is a typical lysosomal enzyme that digests microbial pathogens. AKP is a non-specific phosphate hydrolase that is closely associated with fish immunity [36]. The complement system plays a critical role in the non-specific immune system of fish. LZM plays an important role in the innate immunity of fish, by lysis of bacterial cell wall peptidoglycans and stimulating the phagocytosis of bacteria. C3, a non-specific immune factor, can protect aquatic animals from pathological infection when it is activated [37]. Decreases in levels of these markers would imply a lower immune ability [21,38,39]. In the current study, the activities of ACP and LZM were not significantly affected by MBP inclusion in the diet of juvenile American eel, AKP activity in the serum was affected significantly only in the MBP18 group, and the C3 levels were significantly lower in the MBP12 and MBP18 groups. This indicated that inclusion of MBP at 6% had no negative impact on non-specific immune function, but inclusion levels of MBP at 12% and 18% could impair the immune ability in juvenile American eels.

### 4.3. Effects of FM Being Replaced by MBP on Intestinal Health of Juvenile American Eel

The intestine is the place for nutrient digestion and absorption [40]. Intestinal digestive enzymes, including lipase, amylase, and protease, are secreted by the digestive organs, and play important roles in the process of digestion and absorption of nutrients. Their activities are important indicators for evaluating the digestion, absorption, and utilization of feeds [41]. In the current study, the inclusion of MBP at 12% and 18% led to reductions in the activities of lipase and protease, implying a decreased capacity to digest the nutrients in the intestine of juvenile American eels. No negative effects on enzyme activities were noted in the inclusion level of MBP at 6%. Similarly, the higher levels of dietary MBP led to reductions in lipase activity in the intestines of turbot [17], and the replacement of FM with *Clostridium autoethanogenum* protein decreased the protease activity in largemouth bass [42].

T-AOC is a comprehensive index to measure the antioxidant capacity [43], SOD, CAT, and GSH-Px are oxidative enzymes that play vital roles in the homeostasis system of scavenging oxygen free radicals to reduce the damage of lipid peroxidation [44]. In the present study, the T-AOC level and SOD activity were decreased when dietary MBP levels were 12% or 18%, and the activities of CAT and GSH-Px in the intestine were decreased sharply by 18% MBP inclusion. Additionally, the intestinal MDA level, an indicator of the degree of lipid peroxidation, was significantly increased only in the MBP18 group. Those results suggested that the higher replacement level of FM by MBP could decrease antioxidant ability in the intestine of juvenile American eels. Similarly, Chen et al. [45] reported that the decreased T-AOC level and SOD activity and the increased MDA level were found in the liver of black sea bream (*Acanthopagrus schlegelii*) with an increase in FM replacement by *Clostridium autoethanogenum* protein.

The VH, MT, and microvillus density of intestinal mucosa can directly reflect the digestive and absorption capacity of fish, and they are well-defined indicators of the health status of fish [24,46]. The inclusion of MBP at 6% did not affect VH and MT but led to a decrease in these two parameters in the MBP12 and MBP18 groups. Meanwhile, scanning electron microscopy revealed that the density of intestinal microvillus was gradually lowered with the increasing levels of MBP substitution for FM, and even shed in the MBP18 group. Similar results were also reported in Pacific white shrimp (*Litopenaeus vannamei*) fed a diet with high inclusion levels of MBP [8]. The interesting fact is that MBP could alleviate enteritis induced by soybean meal in the diet of Atlantic salmon [20] and has no negative effect on the intestinal structure of largemouth bass [21]. It seemed that the change in intestinal morphology appears to be related to species as well as the inclusion level of MBP in diets. Further research on how MBP interacts with the intestinal barrier of aquatic animals is worthwhile.

The intestinal microbiota of fish can help to digest and absorb nutrients, regulate immune function, and has a mutually beneficial symbiotic relationship with the host [47]. Microbial dysbiosis in the intestine can lead to metabolic disorders and impact intestinal health [48]. Generally, alpha diversity analysis can reflect the relative abundance and diversity of the microbiota [49]. In this study, the results of Chao1, Observed_species, PD_whole_tree, and Shannon presented no significant difference among all MBP groups, which indicated that the FM replacement level by MBP did not affect the alpha diversity of the intestinal microbiota. Firmicutes and Actinobacteriota are typically noted as beneficial bacteria in the intestine [50,51], while Proteobacteria and Bacteroidota are normally considered opportunistic pathogenic bacteria [8,52]. In this study, the relative abundance of Firmicutes increased first and then decreased with the dietary MBP inclusion level from 0 to 18%, while the changes in relative abundances of Proteobacteria, Actinobacteriota, and Bacteroidota showed the opposite trend. However, compared with the MBP0 group, there was no significant change in the relative abundance of the Firmicutes and Proteobacteria in the intestine of juvenile American eel fed the diet with 18% MBP, while the relative abundance of Actinobacteriota decreased and the relative abundance of Bacteroidota increased. These results suggested that the composition of the intestinal microbiota of American eel might be influenced by MBP.

In the present study, the 6% FM substituted by MBP could increase the relative abundances of *Enterococcus*, *Macrococcus*, *Staphylococcus*, *Clostridium-sensu-stricto*, *Aeromonas,* and *Schlegelella*. Interestingly, *Enterococcus* has a role in promoting digestion in fish [53], and *Macrococcus* was reported to increase nutrient metabolism and growth of fish [54]. *Staphylococcus*, *Clostridium-sensu-stricto*, *Aeromonas,* and *Schlegelella* are opportunistic pathogenic bacteria in the fish intestine that could remain at low, non-pathogenic levels in the fish for a long time with no impact on fish health [55,56]. Therefore, given the relative abundance of the various bacteria mentioned above, the appropriate MBP level appear to promote the proliferation of some beneficial bacteria in the intestine of the juvenile American eel. However, the relative abundances of *Rhodobacter*, *Prevotellaceae_NK3B31_group,* and *Brevundimonas* [57,58,59] were significantly elevated in the MBP18 group, these bacteria were considered to be opportunistic pathogenic bacteria. *Rhodobacter* was noted as being more abundant in the intestine of grass carp (*Ctenopharyngodon idellus*) with intestinal diseases compared with healthy individuals [59]. Similarly, *Prevotellaceae* was reported as the conditional pathogen in soybean meal-induced intestinal inflammation in turbot [60], and the relative abundance of *Brevundimonas* was increased with the increase in dietary starch levels from 5% to 15% in largemouth bass [61]. Those results suggested that the damage noted in the intestinal health at the MBP replacement level of 18% could be partly attributed to the decrease in the relative abundances of some beneficial bacteria and the increase in the proliferation of some potentially pathogenic bacteria, which should be clarified in further studies.

Overall, the results of the present study demonstrated that the inclusion of MBP at 6% in diets of American eels has no impact on a range of growth measures, serum biochemistry, digestive enzymes, antioxidant capacity, or intestinal morphology, and, therefore, can be considered as a suitable replacement for FM in juvenile American eels. However, higher inclusion levels of MBP (12% and 18%) resulted in some negative effects on the growth and health status of the American eel. One possible explanation for the poor growth performance might be a reduction in digestibility at higher inclusion levels of MBP [15], particularly as the various cellular components in these bacterial proteins could be resistant to digestion [13,62]. Additionally, the presence of lipopolysaccharides (LPSs) coated on MBP might have contributed to negatively affecting serum biochemical parameters and intestinal health status in the present study [63]. LPSs were confirmed to induce oxidative stress, suppress digestive and immune status, and damage the intestinal barrier of aquatic animals [8,64]. However, for largemouth bass, the presence of LPSs and copper in the SCPs were noted as positive drivers for health [21] and thus the effect of high MBP inclusion might be species-specific. Furthermore, the special odor of MBP could decrease the feed intake of juvenile American eels in the present study, which could be proved in the FR of the MBP18 group.

## 5. Conclusions

In conclusion, MBP could replace 6% of FM in the diet without affecting the growth performance, biochemical parameters in the serum, and intestinal health of juvenile American eels. High inclusion levels of MBP at 12% and 18% in the diet might exert some negative effects on growth and health status, the specific mechanism needs to be elucidated in further research.

## Figures and Tables

**Figure 1 animals-13-00681-f001:**
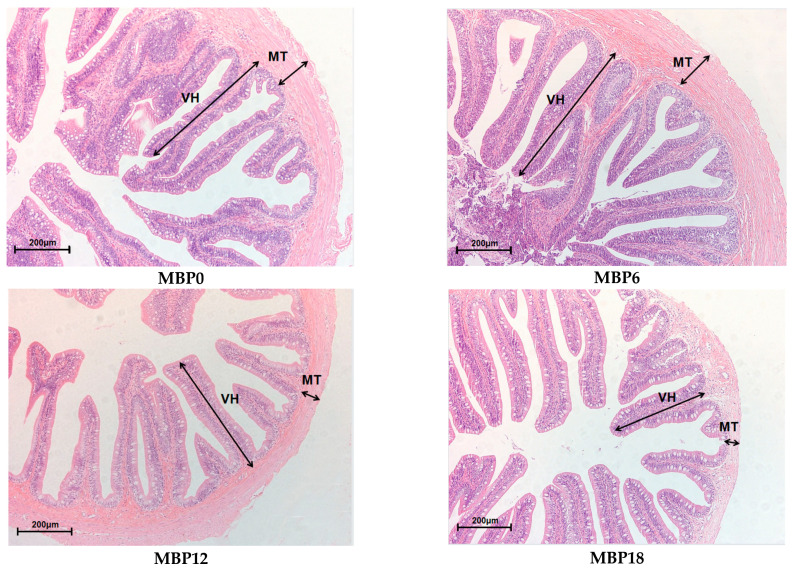
Effects of FM being replaced by MBP on intestinal morphology of the juvenile American eel. VH: villus height; MT: muscular thickness. Magnification ×200.

**Figure 2 animals-13-00681-f002:**
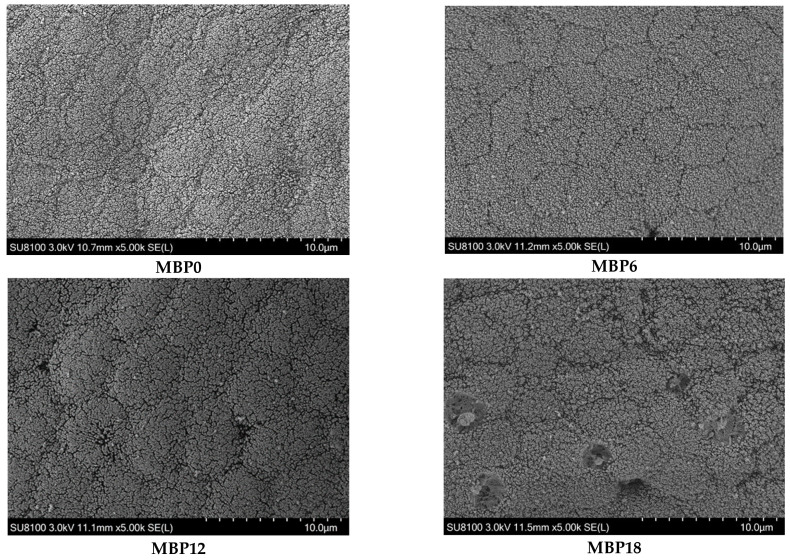
Effects of FM being replaced by MBP on intestinal microvillus of the juvenile American eel. Magnification ×5000. Instrument model: SU8100 (Hitachi Regulus 8100, Hitachi, Tokyo, Japan); Working voltage: 3.0 kV; Working distance (Distance from lens to sample): 11.5 mm; Magnification: 5.00k; Photo mode: SE(L).

**Figure 3 animals-13-00681-f003:**
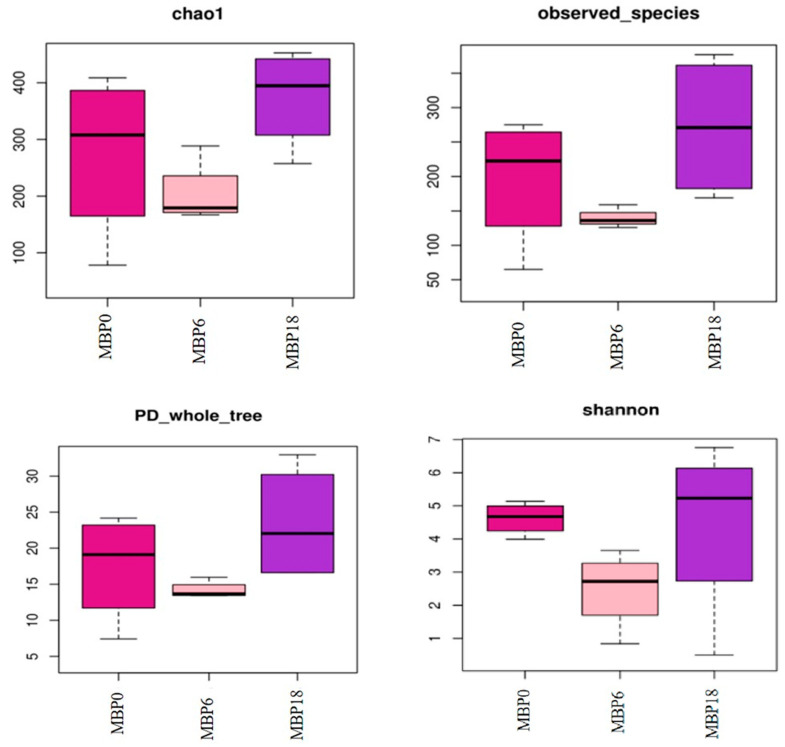
The alpha diversity of intestinal microbiota in the juvenile American eel.

**Figure 4 animals-13-00681-f004:**
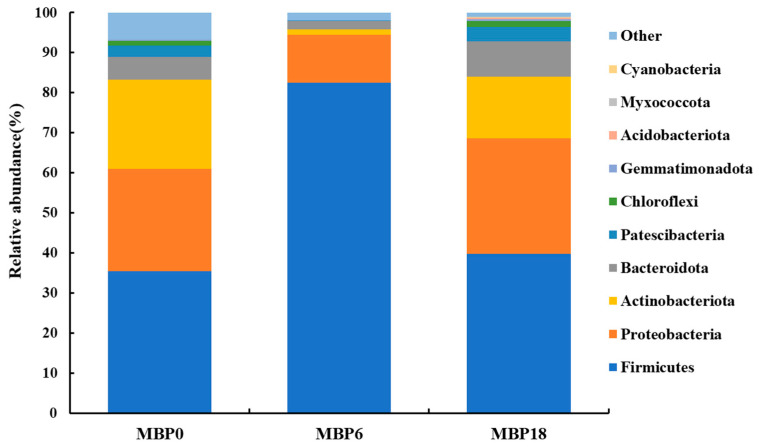
Effects of FM being replaced by MBP on intestinal microbiota composition at phylum level of the juvenile American eel.

**Figure 5 animals-13-00681-f005:**
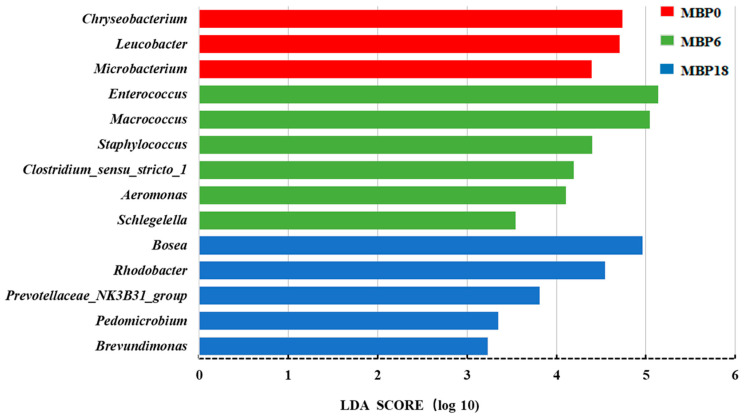
The Lefse analysis in intestinal microflora at genus level of the juvenile American eel. LDA: Linear Discriminant Analysis.

**Table 1 animals-13-00681-t001:** Composition and nutrient levels of trial diets (% dry matter).

Ingredients	Groups			
	MBP0	MBP6	MBP12	MBP18
White fish meal (Super prime) ^a^	40.00	34.00	28.00	22.00
Methanotroph bacteria protein ^b^	0.00	6.00	12.00	18.00
Brown fish meal (Super prime) ^c^	30.00	30.00	30.00	30.00
α-starch	23.50	23.50	23.50	23.50
Puffed soybean meal	2.00	2.00	2.00	2.00
Brewers yeast	2.00	2.00	2.00	2.00
Fish oil	0.00	0.2	0.3	0.4
Choline chloride	0.50	0.50	0.50	0.50
Monocalcium phosphate	0.50	0.50	0.50	0.50
Vitamin premix ^d^	0.40	0.40	0.40	0.40
Mineral premix ^e^	0.60	0.60	0.60	0.60
Microcrystalline cellulose	0.50	0.30	0.20	0.10
Proximate analysis				
Dry matter	92.20	92.25	92.38	92.36
Crude protein	49.24	49.27	49.30	49.34
Crude lipid	5.98	6.03	5.99	5.94
Ash	13.35	13.27	13.07	12.93
Gross energy (MJ/kg)	18.05	18.09	18.14	18.15

^a^ White fish meal (Super prime): 68.89% crude protein and 7.04% crude lipid. It was mainly processed from cod, which was provided by Alaska Coastal Airlines, Washington, USA. ^b^ Methanotroph bacteria protein (*Methylococcus capsulatus*, Bath): 68.40% crude protein, 4.55% crude lipid (FeedKind^®^, Calysta, Inc., CA, USA). ^c^ Brown fish meal (Super prime): 67.92% crude protein, 7.66% crude lipid. It was mainly processed from anchovy, which was provided by Copeinca ASA, Peru. ^d^ Vitamin premix (IU or mg/kg of diet): VA 400,000 IU; VD_3_ 100,000 IU; VE 5000 mg; VK_3_ 1000 mg; VB_1_ 800 mg; VB_2_ 800 mg; VB_6_ 800 mg; VB_12_ 6 mg; VC 15,000 mg; calcium pantothenate 2800 mg; nicotinic acid 5000 mg; folic acid 380 mg; biotin 8 mg; inositol 7000 mg. ^e^ Mineral premix (mg/kg of diet): FeSO_4_·H_2_O 666.67 mg; ZnSO_4_·H_2_O 202.89 mg; MgSO_4_·H_2_O 5094.34 mg; Cu_2_(OH)_3_Cl 12.05 mg; CoSO_4_ 3.64 mg; Na_2_SeO_3_ 0.89 mg; Ca(IO_3_)_2_⋅6H_2_O 2.59 mg; zeolite powder 16.93 mg.

**Table 2 animals-13-00681-t002:** Amino acid levels of trial diets (% dry matter).

Amino Acid	Groups			
	MBP0	MBP6	MBP12	MBP18
Threonine	2.27	2.26	2.25	2.23
Valine	2.60	2.59	2.54	2.52
Methionine	1.40	1.39	1.34	1.31
Isoleucine	2.22	2.23	2.17	2.10
Leucine	3.88	3.87	3.80	3.71
Phenylalanine	1.90	1.91	1.87	1.83
Histidine	1.61	1.68	1.68	1.58
Lysine	4.09	4.00	3.91	3.78
Arginine	2.95	2.92	2.98	2.86
Tryptophan	0.58	0.62	0.67	0.68
Essential amino acid	23.50	23.47	23.21	22.60
Aspartic acid	4.75	4.67	4.59	4.50
Glutamic acid	6.52	6.53	6.19	6.14
Glycine	2.82	2.77	2.76	2.74
Cysteine	0.51	0.51	0.47	0.43
Tyrosine	1.80	1.77	1.80	1.71
Proline	2.02	2.22	2.02	2.11
Serine	2.07	2.04	1.99	1.94
Alanine	3.05	3.09	3.10	3.14
Non-essential amino acid	23.54	23.6	22.92	22.71

**Table 3 animals-13-00681-t003:** Effects of FM being replaced by MBP on growth performance parameters of the juvenile American eel.

Items	Groups			
	MBP0	MBP6	MBP12	MBP18
SR (%)	92.48 ± 1.65	90.82 ± 1.65	92.48 ± 1.65	92.50 ± 4.18
IBW (g/fish)	26.97 ± 0.03	26.99 ± 0.03	26.96 ± 0.02	27.00 ± 0.02
FBW (g/fish)	45.39 ± 1.72 ^a^	45.18 ± 0.80 ^a^	43.74 ± 0.90 ^a^	40.05 ± 1.29 ^b^
WGR (%)	68.25 ± 6.38 ^a^	67.32 ± 2.99 ^a^	62.20 ± 3.35 ^a^	48.34 ± 4.81 ^b^
SGR (%/d)	0.75 ± 0.05 ^a^	0.74 ± 0.03 ^a^	0.68 ± 0.03 ^a^	0.56 ± 0.05 ^b^
FR (%)	1.11 ± 0.02 ^a^	1.09 ± 0.02 ^ab^	1.07 ± 0.03 ^b^	0.99 ± 0.02 ^c^
FE (%)	65.31 ± 5.60 ^a^	65.50 ± 2.54 ^a^	63.57 ± 3.61 ^a^	55.95 ± 5.30 ^b^
PER (%)	130.75 ± 11.22 ^a^	131.14 ± 5.10 ^a^	127.27 ± 7.24 ^a^	112.01 ± 10.62 ^b^

The values are means ± SD, *n* = 4. In the same row, values with different letter superscripts are significantly different (*p* < 0.05). SR: survival rate; IBW: initial body weight; FBW: final body weight; WGR: weight gain rate; SGR: specific growth rate; FR: feeding rate; FE: feed efficiency; PER: protein efficiency ratio.

**Table 4 animals-13-00681-t004:** Effects of FM being replaced by MBP on serum biochemical parameters of the juvenile American eel.

Items	Groups			
	MBP0	MBP6	MBP12	MBP18
GOT (U/L)	5.37 ± 0.34 ^c^	9.56 ± 0.69 ^b^	13.51 ± 0.40 ^a^	13.60 ± 0.87 ^a^
GPT (U/L)	9.02 ± 2.26 ^a^	12.61 ± 0.91 ^b^	13.98 ± 0.47 ^b^	34.38 ± 2.75 ^a^
D-lac (nmol/mL)	8.11 ± 0.64 ^c^	7.99 ± 0.38 ^c^	9.21 ± 0.63 ^b^	12.00 ± 0.02 ^a^
DAO (U/L)	28.43 ± 0.04 ^c^	27.40 ± 0.92 ^c^	29.75 ± 0.72 ^b^	36.27 ± 1.35 ^a^
LZM (U/mL)	9.10 ± 1.79	8.70 ± 0.81	8.77 ± 1.03	7.61 ± 1.29
ACP (U/mL)	9.76 ± 0.38	8.30 ± 1.08	9.40 ± 1.59	10.15 ± 0.94
AKP (U/mL)	1.19 ± 0.14 ^a^	1.02 ± 0.03 ^ab^	1.14 ± 0.05 ^a^	0.93 ± 0.09 ^b^
C3 (μg/mL)	219.86 ± 10.25 ^a^	220.42 ± 11.11 ^a^	110.86 ± 4.03 ^b^	81.70 ± 0.86 ^c^

The values are means ± SD, *n* = 4. In the same row, values with different letter superscripts are significantly different (*p* < 0.05). GOT: glutamic-oxaloacetic transaminase; GPT: glutamic-pyruvic transaminase; D-lac: D-lactate; DAO: diamine oxidase; LZM: lysozyme; ACP: acid phosphatase; AKP: alkaline phosphatase; C3: complement 3.

**Table 5 animals-13-00681-t005:** Effects of FM being replaced by MBP on intestinal digestive enzyme activities of the juvenile American eel.

Items	Groups			
	MBP0	MBP6	MBP12	MBP18
Amylase (U/mg prot)	1.92 ± 0.27	1.98 ± 0.16	2.17 ± 0.22	2.32 ± 0.25
Lipase (U/g prot)	7.19 ± 0.41 ^a^	6.37 ± 0.49 ^a^	5.01 ± 0.94 ^b^	2.73 ± 0.16 ^c^
Protease (U/mg prot)	51.41 ± 1.82 ^a^	49.45 ± 1.94 ^ab^	47.01 ± 2.69 ^b^	34.42 ± 2.11 ^c^

The values are means ± SD, *n* = 4. In the same row, values with different letter superscripts are significantly different (*p* < 0.05).

**Table 6 animals-13-00681-t006:** Effects of FM being replaced by MBP on intestinal antioxidant parameters of the juvenile American eel.

Items	Groups			
	MBP0	MBP6	MBP12	MBP18
T-AOC (mmol/g prot)	0.22 ± 0.03 ^a^	0.17 ± 0.09 ^a^	0.11 ± 0.02 ^ab^	0.06 ± 0.04 ^b^
SOD (U/mg prot)	38.68 ± 2.23 ^a^	38.66 ± 1.66 ^a^	36.03 ± 3.43 ^ab^	33.25 ± 2.09 ^b^
CAT (U/mg prot)	3.40 ± 0.46 ^a^	3.45 ± 0.64 ^a^	3.65 ± 0.28 ^a^	2.71 ± 0.14 ^b^
GSH-Px (U/mg prot)	19.69 ± 0.55 ^a^	18.23 ± 1.15 ^a^	17.90 ± 1.05 ^a^	14.58 ± 1.20 ^b^
MDA (nmol/mg prot)	0.62 ± 0.12 ^b^	0.51 ± 0.03 ^b^	0.66 ± 0.03 ^b^	1.11 ± 0.24 ^a^

The values are means ± SD, *n* = 4. In the same row, values with different letter superscripts are significantly different (*p* < 0.05). T-AOC: total antioxidant capacity; SOD: superoxide dismutase; CAT: catalase; GSH-PX: glutathione peroxidase; MDA: malondialdehyde.

**Table 7 animals-13-00681-t007:** Effects of FM being replaced by MBP on intestinal villus height (VH) and muscular thickness (MT) of the juvenile American eel.

Items	Groups			
	MBP0	MBP6	MBP12	MBP18
VH (µm)	643.01 ± 113.42 ^a^	656.80 ± 78.25 ^a^	525.77 ± 67.54 ^b^	410.50 ± 79.39 ^c^
MT (µm)	169.17 ± 34.02 ^a^	168.36 ± 26.98 ^a^	122.20 ± 20.09 ^b^	94.91 ± 10.67 ^c^

The values are means ± SD, *n* = 4. In the same row, values with different letter superscripts are significantly different (*p* < 0.05).

## Data Availability

The data that support the findings of this study are available from the corresponding author upon reasonable request.

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
