# Peer review of "Evaluation of Methanotroph (Methylococcus capsulatus, Bath) Bacteria Protein as an Alternative to Fish Meal in the Diet of Juvenile American Eel (Anguilla rostrata)"

_animals, 2023, doi:10.3390/ani13040681_

Round 1

Reviewer 1 Report

Dear editor,

Greeting!

On behave of review the manuscript “Evaluation of Methanotroph (Methylococcus capsulatus, Bath) Bacteria Protein as an Alternative to Fish Meal in the Diet of Juvenile American Eel (Anguilla rostrata)” by Wenqi Lu And co-authors.

It is an urgent issue to reduce the consumption of high-quality fish meal for the sustainable development of eel aquaculture. Methanotroph (Methylococcus capsulatus, Bath) bacteria protein is shown to be a promising alternative to fish meal due to its high nutritional values and lower production costs. The experimental design is sound and the results are also reliable. This study provides a first perspective on the feasibility of replacing fish meal with bacteria protein in the diet of American eel. However, several major comments that need to be addressed by authors.

1. In the experimental design, how do 4 kinds of trial diet ensure isonitrogen(about 49% of crude protein) and isofat (about 6% of crude lipid).

2. It is recommended to add a fluorescence quantitative dimension to intestinal antioxidant results.

3. In intestinal microbiota analysis, Why only MBP0, MBP6 and MBP18 samples were selected for intestinal microbiome analysis and not MBP12?

4. L 65-67 “Furthermore, the emerging evidence suggested that MBP might have beneficial effects on the antioxidant capacity, immunity, and intestinal health of some fish species [13-16].” The conclusion of this study is "MBP could successfully replace 6% of FM in the die without adverse effects". It is curious whether MBP has produced some beneficial effects, such as in blood and intestinal health, which can highlight the productive significance of this study.

5. L 277-279 “As evidenced in the SEM view, the intestinal microvillus density in MBP12 and MBP18 groups was significantly sparser than those in MBP0 and MBP6 groups (p < 0.05), there was apparent damage of the intestinal microvillus in the MBP18 group (Figure 2). The density of intestinal microvilli has not been quantified, so the "significantly sparser" described is inappropriate.

Author Response

  1. In the experimental design, how do 4 kinds of trial diet ensure isonitrogen (about 49% of crude protein) and isofat (about 6% of crude lipid).

Re: The protein levels of MBP and fish meal to be replaced were similar, please see the note in table 1. The fish meal was replaced by MBP in equal proportion, so the protein levels of the four diets were similar. The different levels of fish oil were added to the diets with MPB supplementation to achieve the isofat among the four diets.

  1. It is recommended to add a fluorescence quantitative dimension to intestinal antioxidant results.

Re: Many thanks for your advices, we did not conduct the determining the fluorescence quantitative dimension to intestinal antioxidant results in present manuscript. we may study those in future research.

  1. In intestinal microbiota analysis, why only MBP0, MBP6 and MBP18 samples were selected for intestinal microbiome analysis and not MBP12?

Re: MBP0 was the control group. There was no significant difference in growth performance, serum biochemical parameters, and other parameters about intestinal health between the MBP0 group and the MBP6 group. We wanted to demonstrate what changes in the intestinal microbiota might be caused by MBP supplementation, and the MBP6 group was selected. The aforesaid parameters in the MBP18 group were the worst among all groups with MBP supplementation, and the changes in those parameters in the MBP12 group were not obvious as those of the MBP18 group. The MBP18 group was more suitable than the MBP12 group to explore the changes in intestinal microbiota.

  1. L 65-67 “Furthermore, the emerging evidence suggested that MBP might have beneficial effects on the antioxidant capacity, immunity, and intestinal health of some fish species [13-16].” The conclusion of this study is "MBP could successfully replace 6% of FM in the die without adverse effects". It is curious whether MBP has produced some beneficial effects, such as in blood and intestinal health, which can highlight the productive significance of this study.

Re: The positive effects of MBP on the antioxidant capacity, immunity, and intestinal health of some fish species might be related to some bioactive material such as immune polysaccharide, nucleic acids, phospholipid, vitamin B, etc.

  1. L 277-279 “As evidenced in the SEM view, the intestinal microvillus density in MBP12 and MBP18 groups was significantly sparser than those in MBP0 and MBP6 groups (p < 0.05), there was apparent damage of the intestinal microvillus in the MBP18 group (Figure 2). The density of intestinal microvilli has not been quantified, so the "significantly sparser" described is inappropriate.

Re: Many thanks for your suggestion, we modified the description of these results. Please see the revised manuscript.

Reviewer 2 Report

The current paper "Evaluation of Methanotroph bacteria protein as an alternative to fish meal in the diet of juvenile American Eel(Anguilla rostruata)" was well written and provide a first perspective on the feasibility of replacing fish meal with bacteria protein in the diet of American eel.

I have several question to talk with the authors.

1. The introduction part, the description on the Methanotroph is too thin to support the title. Methanotroph is one kind of bacteria protein, and the purpose for the current study is to use one protein to replace one protein. In my personal opinion, the the statement of characteristics of candidate protein is important. The relevant information is lacking in the current manuscript.

2. Why did you choose two kinds of fish meal in the trial diets? The Brown one is no need to replace? 

3. FR and FE express the same result, no need to keep both.

4. I did not check the feed intake values, if the Methanotroph have special smell which could affect the palatability?

5. I did not find about the total bacteria count checked by qPCR?If not, the intestinal microbiota analysis will get less meaningful.

6.

Author Response

  1. The introduction part, the description on the Methanotroph is too thin to support the title. Methanotroph is one kind of bacteria protein, and the purpose for the current study is to use one protein to replace one protein. In my personal opinion, the statement of characteristics of candidate protein is important. The relevant information is lacking in the current manuscript.

Re: Many thanks for your suggestion, we added some description about the characteristics of candidate protein. Please see the revised manuscript.

  1. Why did you choose two kinds of fish meal in the trial diets? The Brown one is no need to replace?

Re: It is the routine use strategy to have two kinds of fish meal in the diets of American eels in China. The nutrients level in the two fish meal are similar, while white fish meal has a higher freshness degree than brown fish meal, which might have more biogenic amine including histamine, and American eel cannot withstand the higher histamine to cause intestinal inflammation and other health problem. The proportion of brown fish meal in the diet was constrained strictly, and the price of brown fish meal is usually lower than that of white fish meal, the difference in the cost might be 3000-4000ï¿¥per ton. It is more meaningful to replace white fish meal than brown fish meal at present.

  1. FR and FE express the same result, no need to keep both.

Re: The FR is the feeding rate, which might reflect the parameter of feed intake, while FE is the feed efficiency to show the feed digestibility.

  1. I did not check the feed intake values, if the Methanotroph have special smell which could affect the palatability?

Re: Yes, the MBP did have a certain special smell. From the change in feeding rate, the low level of MBP (6%) in the diet of American eel did not affect the palatability, but 12% and 18% MBP in the diet significantly lowered the feeding rate. Please see the results of this manuscript.

  1. I did not find about the total bacteria count checked by qPCR? If not, the intestinal microbiota analysis will get less meaningful.

Re: Many thanks for your suggestion. We did not determine the total bacteria count checked by qPCR, the High-throughput sequencing was conducted to demonstrate the alpha diversity and relative richness of intestinal bacteria.

Reviewer 3 Report

This paper “Evaluation of Methanotroph (Methylococcus capsulatus, Bath) 2 Bacteria Protein as an Alternative to Fish Meal in the Diet of Ju-3 venile American Eel (Anguilla rostrata)has a potential to be accepted, but some important queries have to be clarified.

·         The abstract needs to be revised from Line 20-26. No need of explaining the methodology in that detail, brief description of methodology can be given.

·         Although the authors recommend 6% replacement of fish meal with Methylococcus capsulatus, Bath) bacteria protein (MBP), but nothing is discussed about the impact of MBP6 in the abstract.

·         Methodology is very clear.

·         The fish meal was replaced with MBP levels of 0, 6%, 12%, and 18% to prepare the diets for four groups. On the basis of what these concentrations were chosen.

·         After completion of 4 weeks acclimatization period, whether the experimental trials were started immediately or fishes were starved for some time. Starvation break is important otherwise the initial diet given during acclimatization may affect the results of this study.

·         Add a reference for the procedure of anesthesia in Line 126.

·         The results and discussion are well written.  

·         Wether the authors have any plan to try MBP in other cultured fishes as well.

·         English language used is fair, however, it is the responsibility of the authors to make sure that the manuscript should possess similarity index which should be acceptable by the editor as per policies of the journal to avoid any plagiarism related issue that could creep in knowingly or un-knowingly. The authors are suggested to confirm this by employing any reliable plagiarism check program/software.

Author Response

  1. The abstract needs to be revised from Line 20-26. No need of explaining the methodology in that detail, brief description of methodology can be given.

Re: Many thanks for your suggestion. We revised the description of the methodology in the abstract, please see the revised manuscript.

  1. Although the authors recommend a 6% replacement of fish meal with Methylococcus capsulatus, Bath) bacteria protein (MBP), but nothing is discussed about the impact of MBP6 in the abstract.

Re: Many thanks for your suggestion, we added some discussion about MBP6 in the abstract.

  1. Methodology is very clear.

Re: Many thanks for your recognition.

  1. The fish meal was replaced with MBP levels of 0, 6%, 12%, and 18% to prepare the diets for four groups. On the basis of what these concentrations were chosen.

Re: The MBP levels in the trial diets were based on the previous reports about the optimal level of MBP in the diets of fish species of references, and we added related descriptions in the revised manuscript.

  1. After completion of 4 weeks acclimatization period, whether the experimental trials were started immediately or fishes were starved for some time. The starvation break is important otherwise the initial diet given during acclimatization may affect the results of this study.

Re: Yes, it is right that our trial started after the trial fish was starved for 24h.

  1. Add a reference for the procedure of anesthesia in Line 126.

Re: Many thanks for your advice, we added a reference for the procedure of anesthesia.

  1. The results and discussion are well written.

Re: Many thanks for your recognition.

  1. Wether the authors have any plan to try MBP in other cultured fishes as well.

Re: Many thanks for your advice, we did plan to try MBP in Pelteobagrus fulvidraco and Hybrid Sturgeon.

  1. English language used is fair, however, it is the responsibility of the authors to make sure that the manuscript should possess similarity index which should be acceptable by the editor as per policies of the journal to avoid any plagiarism related issue that could creep in knowingly or un-knowingly. The authors are suggested to confirm this by employing any reliable plagiarism check program/software.

Re: Many thanks for your suggestion, we simplify some descriptions of measurement methods, and we have checked the similarity to avoid any plagiarism.

Reviewer 4 Report

In this study, the potential of replacing fishmeal in the diet of juvenile of American Eel (Anguilla rostrata) with a protein of methanotroph bacterial origin (Methylococcus capsulatus, Bath) was investigated. Several concentrations 0, 6, 12 and 18% were selected. Growth and blood serum parameters, intestinal oxidative stress, morphology and intestinal microbiota were investigated. Although, in terms of growth parameters, the differences are small between the different concentrations, the rest of the investigated parameters indicate that negative effects occur at concentrations of 12 and 18%. The study is topical. Specific comments: why didn't you use the MBP0 diet during the acclimatization period? The method does not specify how the intestinal villus height and muscular thickness of juvenile was measured, please specify.

Why in the graphs in figure 3 and 4 only 3 experimental variants are referred to and not 4? Please specify

Author Response

  1. Specific comments: why didn't you use the MBP0 diet during the acclimatization period?

Re: During the acclimatization period, all eels were fed on a commercial diet manufactured by Fuzhou Xinruiyi Industrial Co., Ltd., Fuzhou, China. Please see the manuscript. The diet formula of MBP0 group is similar with that of the commercial diet for American eel with higher inclusion of white fish meal.

  1. The method does not specify how the intestinal villus height and muscular thickness of juvenile was measured, please specify.

Re: Many thanks for your suggestion, we added a reference to specify the measurement of those parameters. Please see the revised manuscript.

  1. Why in the graphs in figure 3 and 4 only 3 experimental variants are referred to and not 4? Please specify

Re: Compared to the MBP0 group, there were no significant changes in growth, serum biochemical parameters, digestive enzymes, and intestinal antioxidant parameters in the MBP6 group, however, these parameters were significantly affected in the MBP12 and MBP18 groups (except for some parameters of the growth in the MBP12 were not affected). Due to the intestinal microbiota has certain effects on fish growth, serum biochemical parameters, and antioxidant aspects, this study aimed to investigate the effects of appropriate and excessive inclusion of MBP on the intestinal microbiota of eel, so the MBP0, MBP6, and MBP18 were selected. To make it clear, we have added this information in the revised manuscript.